# Crashworthiness Analysis and Multi-Objective Optimization for Concave I-Shaped Honeycomb Structure

**Tingting Wang, Mengchun Li, Dongchen Qin \*, Jiangyi Chen and Hongxia Wu**

School of Mechanical and Power Engineering, Zhengzhou University, Zhengzhou 450001, China
\* Correspondence: dcqin@zzu.edu.cn; Tel.:+86-13623868575

**Abstract:** Due to their superior structural and mechanical properties, materials with negative Poisson's ratio are of increasing interest to research scholars, especially in fuel-efficient vehicles. In this work, a new concave I-shaped honeycomb structure is established by integrating the re-entrant hexagon and the I-shaped beam structure, and its negative Poisson's ratio characteristics and energy absorption properties are investigated. The effect of structural parameters on the energy absorption characteristics is analyzed using the finite element model. The results show that both the specific energy absorption and peak impact force decrease with the increase in cellular length and vertical short cellular height, and increase with the increase in horizontal short cellular length and cellular thickness. To obtain a smaller peak impact force and larger specific energy absorption with smaller mass, the four cell sizes were optimized by using Latin hypercube sampling, Gaussian radial basis function, and non-dominated sorting genetic algorithm II (NSGA-II). Compared with the original design, the SEA increased by 44.175%, and the PCF increased by 25.857%. Meanwhile, the mass decreased by 31.140%. Hence, the optimal structure has better crashworthiness.

**Keywords:** honeycomb structure; negative Poisson's ratio; crashworthiness; multi-objective optimization; finite element

## 1. Introduction

Materials with positive Poisson's ratio expand laterally when subjected to axial pressure and contract laterally when subjected to axial tension. In contrast, materials with negative Poisson's ratio compress laterally under axial pressure. These materials were first studied in 1944 by the scientist Love [1], who observed a Poisson's ratio of −0.14 while studying the mechanical deformation behavior of a natural pyrite cubic single crystal structure and found that it had a significant internal shrinkage effect. Later on, dozens of natural negative Poisson's ratio crystal structures were discovered one after another, and negative Poisson's ratio materials were widely studied and applied.

Evans [2] referred to these substances and structures as auxetics. Because of this result, materials with negative Poisson's ratio are also called pull-up materials. The equivalent modulus of elasticity was researched by Gibson [3], beginning with material deformation. Robert [4] modified the size and angle of the typical re-entrant hexagon structure to make Poisson's ratio −1. Herakovich [5] investigated the impact of transverse cracking on the elastic modulus and Poisson's ratio using both analytical approximation results and finite element technique solutions. Horrigan [6] optimized the structure of materials with a negative Poisson's ratio to improve their energy absorption abilities. When Liu et al. [7] added irregularity to the negative Poisson's ratio honeycomb structure, they discovered that the attributes of the negative Poisson's ratio under high-speed impact were accelerated. To confirm that the theoretical derivation of the negative Poisson's ratio equations matched the numerical results, Wang et al. [8] investigated the dynamic response of a novel type of negative Poisson's ratio honeycomb structure with multiple internal concave angles under in-plane impacts. By merging the circular element with the folded plates, Wu et al. [9]

created a cross-circular honeycomb model. The deformation mode of the cross honeycomb has a zero Poisson's ratio and a strong energy absorption capability.

In recent years, the research aimed at developing a material model which is equivalent to the mechanical response of a lattice structure has provided a good reference for how to calculate the equivalent properties. In [10], the mechanical behavior of a body-centered cubic (BBC) configuration under compression and within the elastic limit is considered. The finite element analysis approach and theoretical calculations are used on a single unit cell BBC for several cases to predict equivalent solid properties. The results are then used to develop a neural network model so that the equivalent solid properties of a BCC lattice of any configuration can be predicted. In [11,12], for the aim of capturing the behavior of the entire heterogeneous cellular structure based on the analyses of the InsideBCC unit cell, only the periodic boundaries would suffice. Regarding both the elastic modulus and Poisson's ratio, the lattice unit cell faces are free to move, including all translational and rotational degrees of freedom for rigid plates. These articles provide an idea on how to calculate the equivalent properties and how to investigate the size of the specimens.

Cellular materials and constructions with negative Poisson's ratio fall under the category of porous materials and have qualities such as low weight, high energy absorption, high sound insulation, and high damping. Scholars have thoroughly analyzed the deformation process of cellular structures, and the present research on cellular structures has increasingly moved from two-dimensional to three-dimensional [13]. Zhang et al. [14] tested the mechanical properties of several cellular architectures and ran dynamic simulations using additive manufacturing technology.

After nearly fifty years of development, materials with negative Poisson's ratio have penetrated micro and macro scales of different sizes and are used in a wide range of applications in military medicine, life, and vehicle safety. The molecular level is in the range of $10^{-7}$ to $10^{-5}$ mm with some crystalline structures and liquid polymorphs. The microscopic sizes from $10^{-4}$ to $10^{-1}$ mm include a wide range of composites and ceramics, as well as metallic foams, honeycomb materials, etc. The macroscopic level is mainly negative Poisson's ratio multi-cell structures with sizes between $10^{-4}$ and 10 mm. The cell structure has obvious periodicity. Thus, the different sizes of the material structure have different ranges of applications, and their potential research technology value for industries and national defense and security has very far-reaching significance [15]. Today's negative Poisson's ratio material structures mainly include concave structures [16], chiral or anti-chiral structures [17], switchable architected structures [18], and origami and fold structures [19].

The traditional concave hexagonal honeycomb structure has been modified by many researchers into a new negative Poisson's ratio structure. These researchers theoretically calculated the relationship between the equivalent elastic modulus and Poisson's ratio of these new structures with each structural parameter. Then, they used simulation software to analyze and compare the analytical and finite element solutions to determine the effects of these structural parameters on the mechanical system. Jiang et al. [20] proposed a novel ring-shaped negative Poisson's ratio structure and investigated the impact of its structural dimensions on the equivalent elastic modulus and negative Poisson's ratio using the design approach of mechanical metamaterials. To determine the link between the equivalent elastic modulus, Poisson's ratio, and each structural parameter, Wang et al. [21] developed a negative Poisson's ratio structure made up of circular curves.

Global automobile ownership is continuously rising along with the industry's fast expansion. According to the World Health Organization's "Global Report on Road Safety 2015", 1.25 million persons died in road accidents globally in 2013 [22], and approximately half of these victims were pedestrians. Studies have shown that pedestrians are more likely to suffer head and lower limb injuries in a collision with a car; while lower limb injuries are generally not fatal, head injuries are up to 62% more likely to be fatal [23]. Therefore, the material and structure of energy-absorbing devices for cars must be strong and lightweight but also have a good energy-absorption effect. When compared to other materials of the

same mass, the negative Poisson's ratio material has a greater energy absorption effect, making it ideal for the investigation of protective devices for human body protection.

Furthermore, there is more and more research on the multi-objective optimization of a cellular structure with a negative Poisson's ratio. By employing the sampling method, which is a uniform experimental design, Liu et al. [24] were able to obtain the fitting equations for the four evaluation indices of energy, specific energy absorption, peak crushing force, and equivalent Poisson's ratio for the geometrical parameters. They optimized the design based on the fitting equations. A metamodel-based optimization method, integrating finite element simulation, kriging metamodel, and multi-objective genetic algorithm, was employed to optimize the design of the hierarchical honeycomb by Yin [25]. The optimization process aimed to achieve the maximum value of specific energy absorption and minimum value of peak crushing force. The majority of academics will use the surrogate model technique for highly nonlinear situations owing to the computational expense. Using nonlinear functions and collision simulation elements, Fang et al. [26] investigated the applicability of the response surface approximation model and the Gaussian radial basis function model. For collision situations, the Gaussian radial basis function model provides a better solution than the response surface model, although it still has drawbacks such as high processing costs.

Considering the discussed issues, a novel concave I-shaped honeycomb structure is proposed in this study based on integrating a conventional re-entrant hexagon structure and I-beam construction. The structure can be interlocked and connected by its cellular units, and its negative Poisson's ratio property is also remarkable. The rest of this article is organized as follows: The geometric configuration and the equivalent Poisson's ratio of a novel cellular structure are established in Section 2. Section 3 analyzes the influence of cell structure with different parameters on the performance indexes. The design optimization method and the optimization results are provided in Section 4. Some conclusions are given in Section 5.

## 2. Related Theories on the Concave I-Shaped Honeycomb Structure

The concave I-shaped honeycomb structure proposed in this paper is an optimized modification of the conventional concave hexagonal honeycomb structure in Figure 1. The traditional re-entrant hexagon honeycomb structure was the first concave honeycomb structure to be created. Its two side edges have a "> <" shape, which is an important structure for impact resistance. To further improve its ability to absorb energy, the re-entrant hexagon structure is combined with the I-beam structure shown in Figure 1 to form a concave I-shaped honeycomb structure, which is a new honeycomb structure with a negative Poisson's ratio. Compared with the traditional re-entrant hexagon honeycomb structure, the horizontal and vertical short cell walls of the concave I-shaped honeycomb will have a good cushioning effect. The relative density of the concave I-shaped honeycomb structure is higher than that of the traditional concave hexagonal honeycomb for the same cell parameters, which gives it better energy absorption characteristics.

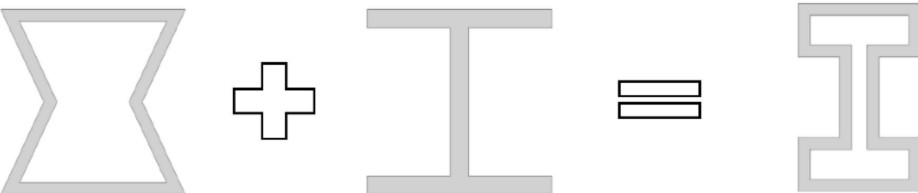

**Figure 1.** Recessed I-shaped honeycomb structure inspired by the combination of the traditional recessed hexagon and I-beam.

### 2.1. Theoretical Calculations

The cellular structure is centrosymmetric, as shown in Figure 2. The cellular length $S$, the cellular height $L$, the vertical short cellular height $h$, the horizontal short cellular length $b$,

and the cellular thickness $t$ all contribute to the cellular structure. The relationship $L = 4 \times h$ must remain constant to guarantee the combined link between cells. A load is given to the cellular structure in the y-direction to investigate the deformation of the cellular structure and the influence of each parameter on the structure's deformation and Poisson's ratio. Since the cellular structure is centrosymmetric, as illustrated in Figure 3, one-fourth of the structure can be used for the investigation. A bending moment of unknown magnitude $M_0$ is applied to the whole structure, while point B is subjected to a force $F/2$ in the y-direction. At the free end E, the corner deformation coordination condition is satisfied.

$$\delta_{11} M_{R1} + \delta_{1F} = 0, \tag{1}$$

where $\delta_{1F}$ is the angle of rotation of point E when only the load $F/2$ is applied and $\delta_{11}$ is the angle of rotation of point E when a unit bending moment is applied at point E.

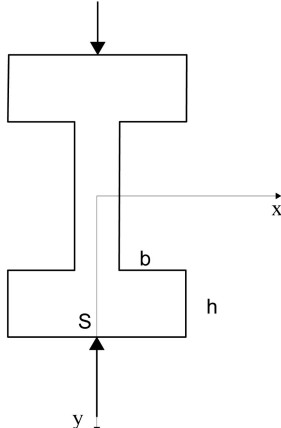

**Figure 2.** The overall cellular structure under load.

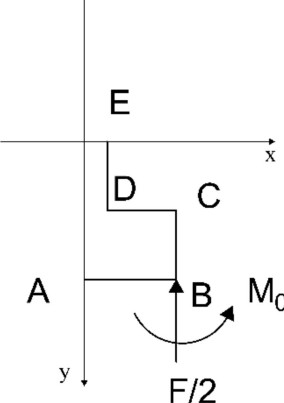

**Figure 3.** The internal forces of a 1/4 structure.

The bending moments that are applied to each of the sections *AB*, *BC*, *CD*, and *DE* when just $F/2$ is used are shown in Figure 4. Figure 5 shows that when acting only on the unit bending moment, the bending moment of each section can be found, and using Mohr's theorem [27] to perform Mohr integral on each beam, the $\delta_{1F}$ and $\delta_{11}$ can be found.

$$M_{AB} = \frac{F}{2} x_1 \quad x_1 \in \left(0, \frac{S}{2}\right) \quad \overline{M_{AB}} = -1, \tag{2}$$

$$M_{BC} = 0 \quad x_2 \in (0, h) \quad \overline{M_{BC}} = -1, \tag{3}$$

$$M_{CD} = \frac{F}{2} x_3 \quad x_3 \in (0, b) \quad \overline{M_{CD}} = -1, \tag{4}$$

$$M_{DE} = \frac{Fb}{2} \quad x_4 \in (0, h) \quad \overline{M_{DE}} = -1, \tag{5}$$

$$\delta_{1F} = \int_0^{\frac{S}{2}} \frac{M_{AB}(x_1)\overline{M_{AB}}(x_1)}{E_m I_m} dx_1 + \int_0^h \frac{M_{BC}(x_2)\overline{M_{BC}}(x_2)}{E_m I_m} dx_2 + \int_0^b \frac{M_{CD}(x_3)\overline{M_{CD}}(x_3)}{E_m I_m} dx_3 + \int_0^h \frac{M_{DE}(x_4)\overline{M_{DE}}(x_4)}{E_m I_m} dx_4, \tag{6}$$

$$\delta_{11} = \int_0^{\frac{S}{2}} \frac{\overline{M_{AB}}(x_1)\overline{M_{AB}}(x_1)}{E_m I_m} dx_1 + \int_0^h \frac{\overline{M_{BC}}(x_2)\overline{M_{BC}}(x_2)}{E_m I_m} dx_2 + \int_0^b \frac{\overline{M_{CD}}(x_3)\overline{M_{CD}}(x_3)}{E_m I_m} dx_3 + \int_0^h \frac{\overline{M_{DE}}(x_4)\overline{M_{DE}}(x_4)}{E_m I_m} dx_4, \tag{7}$$

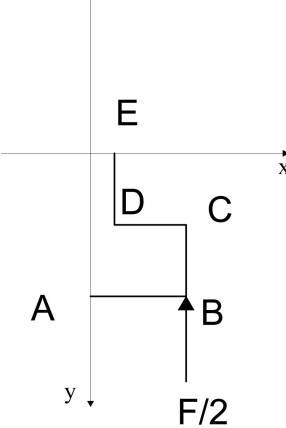

**Figure 4.** One-fourth structure when only $F/2$ load is applied.

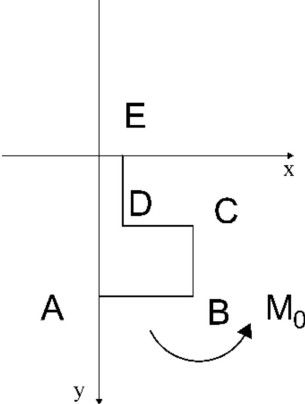

**Figure 5.** One-fourth structure when acting only on unit bending moment.

In Equations (6) and (7), $E_m$ is the modulus of elasticity of the material; $I_m$ is the moment of inertia of the section for the neutral axis.

From Equations (6) and (7), $M_0$ and $\alpha$ are calculated as

$$M_0 = -\frac{\delta_{1F}}{\delta_{11}} = \frac{F}{8}\left(\frac{S^2 + 4b^2 + 8bh}{S + 4h + 2b}\right) = F\alpha, \tag{8}$$

$$\alpha = \frac{1}{8}\left(\frac{S^2 + 4b^2 + 8bh}{S + 4h + 2b}\right), \tag{9}$$

Using the Castigliano theorem [28], the displacement deformation of each segment in the y-direction under a symmetrical load $F/2$ in the y-direction can be found:

$$M_{AB} = \frac{F}{2}x_1 - M_0 \quad x_1 \in \left(0, \frac{S}{2}\right) \quad \overline{M_{AB}} = x_1 - 2\alpha, \tag{10}$$

$$M_{BC} = -M_0 \quad x_2 \in (0,h) \quad \overline{M_{BC}} = -2\alpha, \tag{11}$$

$$M_{CD} = \frac{F}{2}x_3 - M_0 \quad x_3 \in (0,b) \quad \overline{M_{CD}} = x_3 - 2\alpha, \tag{12}$$

$$M_{DE} = \frac{Fb}{2} - M_0 \quad x_4 \in (0,h) \quad \overline{M_{DE}} = b - 2\alpha, \tag{13}$$

$$\Delta_{YY} = \frac{2}{E_m I_m}(\int_0^{\frac{S}{2}} M_{AB}(x_1)\overline{M_{AB}}(x_1)dx_1 + \int_0^h M_{BC}(x_2)\overline{M_{BC}}(x_2)dx_2+ \\ \int_0^b M_{CD}(x_3)\overline{M_{CD}}(x_3)dx_3 + \int_0^h M_{DE}(x_4)\overline{M_{DE}}(x_4)dx_4), \tag{14}$$

$$\Delta_{YY} = \frac{F}{24E_m I_m}(S^3 - 12\alpha S^2 + 48\alpha^2 S + 192\alpha^2 h + 8b^3 \\ -48\alpha b^2 + 96\alpha^2 b + 24b^2 h - 96\alpha bh), \tag{15}$$

As shown in Figure 6, the unit force method can be used to find the displacement deformation of each segment in the x-direction under a symmetrical load $F/2$ in the y-direction.

$$M_{AB} = \frac{F}{2}x_1 - M_0 \quad x_1 \in \left(0, \frac{S}{2}\right) \quad \overline{M_{AB}} = 0, \tag{16}$$

$$M_{BC} = -M_0 \quad x_2 \in (0,\text{h}) \quad \overline{M_{BC}} = x_2, \tag{17}$$

$$M_{CD} = \frac{F}{2}x_3 - M_0 \quad x_3 \in (0,b) \quad \overline{M_{CD}} = h, \tag{18}$$

$$M_{DE} = \frac{Fb}{2} - M_0 \quad x_4 \in (0,h) \quad \overline{M_{DE}} = x_4 + h, \tag{19}$$

$$\Delta_{YY} = \frac{2}{E_m I_m}(\int_0^{\frac{S}{2}} M_{AB}(x_1)\overline{M_{AB}}(x_1)dx_1 + \int_0^h M_{BC}(x_2)\overline{M_{BC}}(x_2)dx_2+ \\ \int_0^b M_{CD}(x_3)\overline{M_{CD}}(x_3)dx_3 + \int_0^h M_{DE}(x_4)\overline{M_{DE}}(x_4)dx_4), \tag{20}$$

$$\Delta_{YX} = \frac{F}{2E_m I_m}\left(3bh^2 - 8\alpha h^2 + hb^2 - 4\alpha hb\right), \tag{21}$$

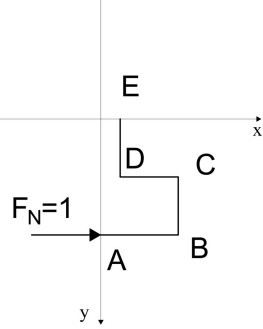

**Figure 6.** Only horizontal unit forces acting on 1/4 of the structure.

The stresses and strains in the y-direction are

$$\sigma_y = \frac{F}{Sz}, \tag{22}$$

$$\varepsilon_y = \frac{\Delta_{YY}}{L}, \tag{23}$$

The Poisson's ratio in the y-direction is

$$\nu_{yx} = -\frac{\Delta_{YX}}{\Delta_{YY}}\frac{L}{S} = -\frac{3bh^2 - 8\alpha h^2 + hb^2 - 4\alpha hb}{S^3 - 12\alpha S^2 + 48\alpha^2 S + 192\alpha^2 h + 8b^3 - 48\alpha b^2 + 96\alpha^2 b + 24b^2 h - 96\alpha bh}\frac{48h}{S}, \tag{24}$$

From the above equations, it can be concluded that Poisson's ratio of the cellular structure in the y-direction, over a small range of deformations, is related to the parameters $S$, $h$, and $b$ of the structure's material.

### 2.2. Comparison of Analytical and Simulation Solutions

Finite element software is used to evaluate the concave I-shaped honeycomb structure, and the findings are compared to the analytical solution in Section 2.1. $S$, $h$, and $b$ are used to examine the laws of the impact of geometrical parameters on the equivalent Poisson's ratio. Figure 7 displays the outcome.

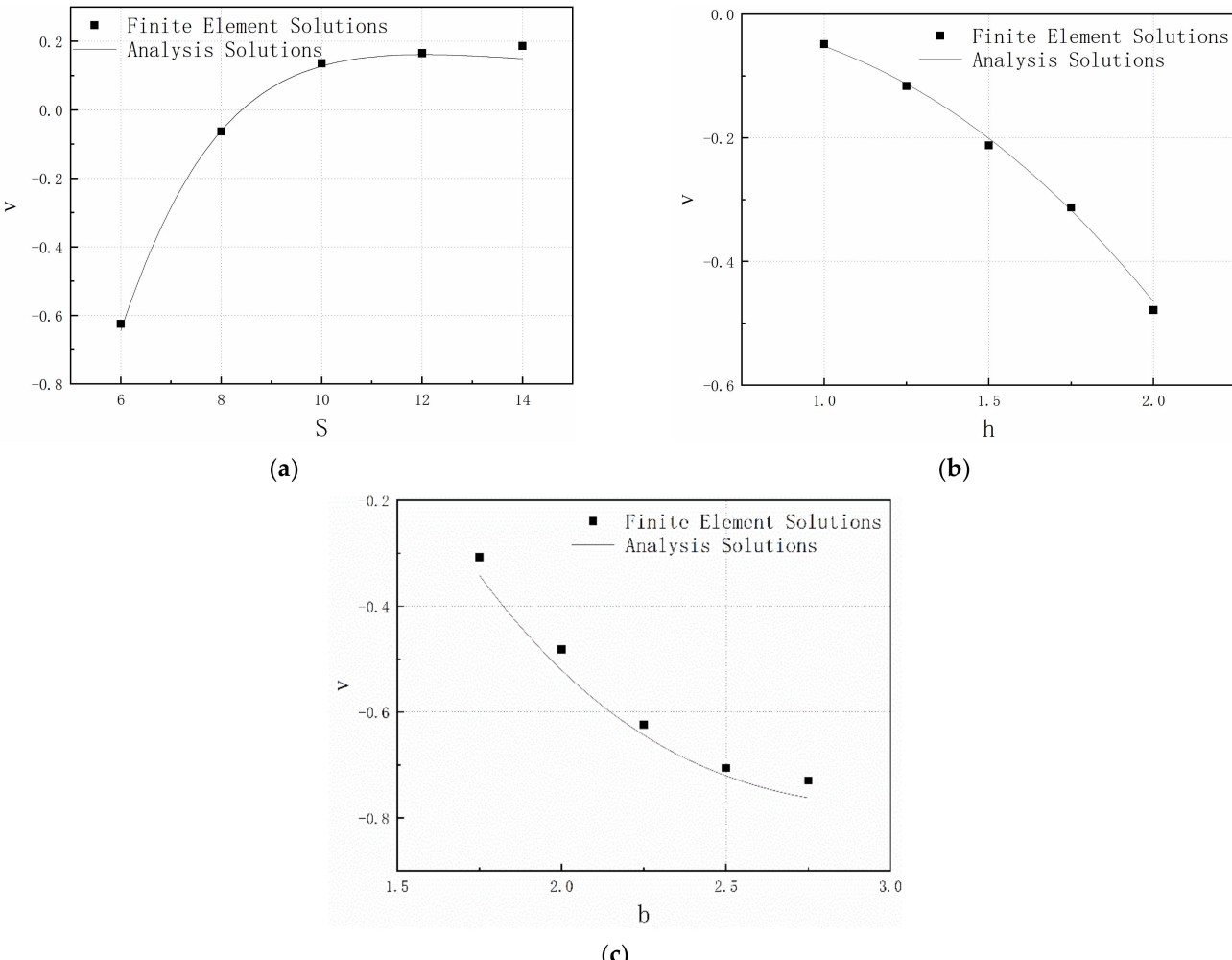

**Figure 7.** Comparison of simulated and analytical solutions when loaded in the y-direction: (**a**) effect of cellular length $S$ on Poisson's ratio; (**b**) effect of vertical short cellular height $h$ on Poisson's ratio; (**c**) effect of horizontal short cellular length $b$ on Poisson's ratio.

The analytical solution expands as $S$ rises, and as $S$ rises higher, the tendency of its growth slows. The error between them becomes reduced as $S$ increases, but as $S$ continues to increase, the error between the analytical solution and the simulated solution becomes worse.

The analytical solution becomes smaller as $h$ rises. The same is true for the simulated solution, where there is little difference between it and the analytical solution, and the outcomes are consistent with those of the analytical equation.

For parameter $b$, the simulated and analytical solutions fall faster at first, but as $b$ increases, they level off.

There are two basic categories of causes for these inaccuracies in the analytical and simulation solutions. One is that the theoretical derivation uses the energy method, which leads to certain mistakes in the findings since tensile and shear force effects are not taken into account during theoretical computation. The second is that because the simulation model's segments are all horizontal or vertical, the force acting on some of them is smaller during theoretical analysis, resulting in smaller segmental integration results. However, because the segments are elastic during the simulation process, they will bend and deform, causing the simulation solution in some cases to be larger than the analytical solution.

From the discussion above, it is clear that when $b$ and $h$ take particular values, an increase in $S$ will also cause the structure to change from a negative Poisson's ratio to a positive Poisson's ratio. When $S$ is taken as a constant value, an increase in $b$ and $h$ can increase the visibility of the structure's negative Poisson's ratio characteristic. By altering the dimensional parameters of the cellular element structure, it is possible to change the Poisson's ratio of the concave I-shaped structure, and the findings provide some theoretical support for additional research.

## 3. Materials and Methods

### 3.1. Finite Element Model

The finite element model is employed to investigate the influence of the energy absorption characteristics and peak crushing force of four cellular parameters systematically. For honeycomb impact resistance studies, the number of cells in the transverse and longitudinal directions is required to be more than nine, so the finite element mode shape in this paper uses a $10 \times 10$ array of periodic cell structures. The concave I-shaped honeycomb structure which uses a commonly used aluminum alloy with density $\rho = 2.7 \times 10^{-9} \text{t}/\text{mm}^3$, elastic modulus $E = 7 \times 10^4 \text{MPa}$, Poisson's ratio $v = 0.33$, and yield stress $\sigma_y = 76 \text{MPa}$ is between the two impact plates. Assuming that the overall member satisfies the Mises reliability criterion and the ideal elastoplastic requirements, the specimen cell element is the S4R shell unit. Two steel plates, 240 mm long, 30 mm wide, and 3 mm thick, one fixed as a base plate and one as a velocity impact plate, are used at a distance of 1 mm from the cellular structure. These two plates are made of steel material with density $\rho = 7.8 \times 10^{-9} \text{t}/\text{mm}^3$, Young's modulus $E = 2.07 \times 10^5 \text{MPa}$, and Poisson's ratio $v = 0.3$. In the software, these two steel plates are endowed with rigid body attributes, and the unit attributes are selected as R3D4. To balance the computing time and accuracy, the cell mesh size of the honeycomb structure is 0.5 mm, and the cell mesh size of the steel plates is 1 mm. Five integration points are taken along the thickness direction of the honeycomb structure to ensure convergence of the results and to reduce the computational effort. The contact type is defined as general contact, and the friction coefficient is taken as 0.02 to simulate the frictional contact between the components of the model during the impact process. The model is shown in Figure 8.

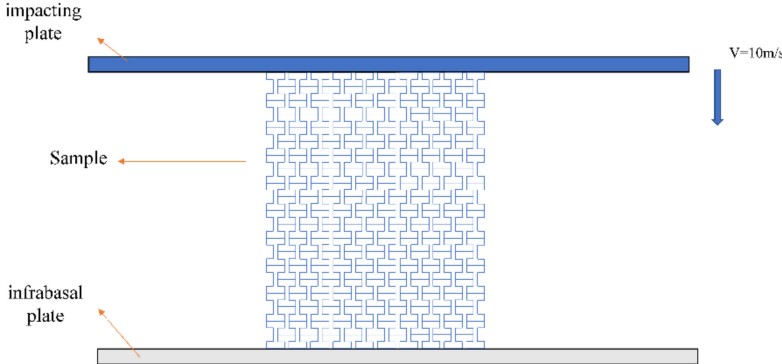

**Figure 8.** Impact on negative Poisson's ratio honeycomb core materials.

A particular velocity is applied to the velocity impact plate. The honeycomb sandwich reduces the energy of the impact on the base plate through the deformation. For the inves-

tigation of the effects of different geometric parameters, the origami-inspired honeycomb sandwich panels were grouped as group A, group B, group C, and group D. Each group consists of five samples, as shown in Table 1. The influence of these parameters on specific energy absorption and peak crushing force is studied. Nowadays, lightweight materials are sought, and the mass should also be considered.

**Table 1.** Characteristics of the investigated concave I-shaped honeycomb structures.

| Sample | Velocity (m/s) | Origami-Inspired Honeycomb | | | |
| --- | --- | --- | --- | --- | --- |
| | | $S$ (mm) | $h$ (mm) | $b$ (mm) | $t$ (mm) |
| A1 | 10 | 6 | 2.25 | 2.25 | 0.65 |
| A2 | 10 | 8 | 2.25 | 2.25 | 0.65 |
| A3 | 10 | 10 | 2.25 | 2.25 | 0.65 |
| A4 | 10 | 12 | 2.25 | 2.25 | 0.65 |
| A5 | 10 | 14 | 2.25 | 2.25 | 0.65 |
| B1 | 10 | 6 | 2.00 | 2.25 | 0.65 |
| B2 | 10 | 6 | 2.25 | 2.25 | 0.65 |
| B3 | 10 | 6 | 2.50 | 2.25 | 0.65 |
| B4 | 10 | 6 | 2.75 | 2.25 | 0.65 |
| B5 | 10 | 6 | 3.00 | 2.25 | 0.65 |
| C1 | 10 | 6 | 2.25 | 1.75 | 0.65 |
| C2 | 10 | 6 | 2.25 | 2.00 | 0.65 |
| C3 | 10 | 6 | 2.25 | 2.25 | 0.65 |
| C4 | 10 | 6 | 2.25 | 2.50 | 0.65 |
| C5 | 10 | 6 | 2.25 | 2.75 | 0.65 |
| D1 | 10 | 6 | 2.25 | 2.25 | 0.50 |
| D2 | 10 | 6 | 2.25 | 2.25 | 0.65 |
| D3 | 10 | 6 | 2.25 | 2.25 | 0.80 |
| D4 | 10 | 6 | 2.25 | 2.25 | 0.95 |
| D5 | 10 | 6 | 2.25 | 2.25 | 1.10 |

*3.2. Experimental Validation*

To verify the accuracy of the finite element model, a test model was built. The test pieces were made by additive manufacturing technology, also known as 3D printing. The model designed in SolidWorks software was saved in stereolithography (STL), which is a format used to define the sample geometry of the 3D printer software. The printer nozzle temperature of 300 °C and the chamber temperature of 77 °C were maintained and used as default temperature settings for all the specimens that were printed. The material we chose is a traditional resin material with high dimensional accuracy, the capacity to achieve fine measurements, and good surface quality for prototypes. As a result, it is employed in many different industries, including the mechanical and automotive sectors. The photosensitive resin 9400e is a more popular 3D printing material since metallic aluminum produces poor results. It has a density $\rho = 1.18 \times 10^{-9} \text{t/mm}^3$, elastic modulus $E = 2.1 \times 10^3 \text{MPa}$, and Poisson's ratio $\nu = 0.35$. The cellular structure was $5 \times 5$ with a single cell structure of $S = 8$ mm, $h = 2.5$ mm, $b = 2.25$ mm, and $t = 0.65$ mm. Its overall length was 54 mm, height was 50 mm, and width was 30 mm. The dimensions of the 3D printed specimen had some errors, but the overall size was similar to that of the test model. A Changchun Kexin WDW-300 microcomputer-controlled electronic universal testing machine, as shown in Figure 9, was used for this test; this machine is mainly used for tensile, compression, and bending tests of metal and non-metal materials. The specimen was placed on the fixed base plate at the bottom of the uniaxial hydraulic press, and the upper compression plate was adjusted to a position 1 mm away from the specimen in advance. The material properties of the specimen and the test program were inputted at the computer end, where the compression speed was 2 mm/min, and the compression displacement was set to 20 mm. Since the pore wall is the main energy-absorbing structure, it plays the main supporting role when subjected to compression. When the structure starts to compress,

the lateral sides of the cytosol below the impact end curl first, and both sides contract to the inside. With the continuous downward pressure of the impact plate, the shear zone of the impact end is continuously driven by the curling deformation of the surrounding cytosol, and the local shear deformation range is continuously extended in the direction of stiffness softening, and the structure will produce an obvious "necking" phenomenon. With the mutual yielding deformation of adjacent cell elements, the cell wall gradually collapses, the material is in full contact, and the structure enters the compact stage. When the cell wall of the structural cell element collapses, the material enters the failure form. A camera was used to record the test process, and a comparison of the simulated and observed compression images is shown in Figure 10.

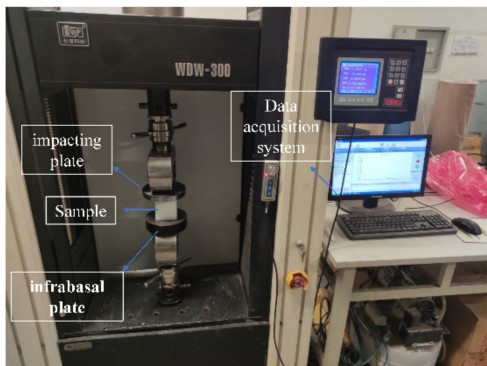

**Figure 9.** Test equipment.

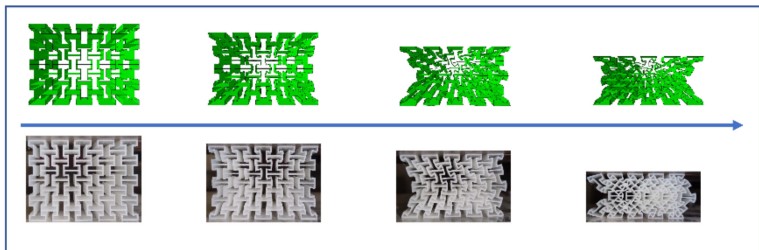

**Figure 10.** Structural compression processes: simulation and experimentation.

Figure 11a shows a comparison of the force–displacement curves for the test solutions and the simulation results. In the elastic stage, the force of the simulation solution and the experimental solution both grow properly, but the force does not considerably increase in the yield stage. The force and displacement both quickly rise during the strengthening phase. Figure 11b shows the stress–strain curves. The linear elastic stage, where the stress and strain largely vary linearly with the slope of the modulus of elasticity, is defined as strain 0–0.1. The stage of the plastic plateau is strain 0.1–0.25. In this stage, there is a clear plateau phase of stress–strain; nonetheless, the overall trend is progressively growing. As the stain grows, the stress varies up and down around the plateau stress. When the strain is more than 0.25, the material enters the compact stage, when it starts to compact and the matrix primarily provides the stress, which causes the stress to climb sharply as the strain increases. Different peaks and variations may be seen on the force–displacement curves produced by the simulation solutions. The total amount of absorbed energy is not significantly impacted by these differences [29]. The average forces in the simulation and test were 1669 N and 1649 N, respectively, with an acceptable error of 1.213%. The slope of the simulation is steeper than that of the test in the elastic stage, which may be caused by flaws in the 3D printed samples and results in a comparatively low force that can be borne, according to a comparison of simulation and test data. The quantity of postprocessing data of the simulation solution and the finite element model's mesh accuracy both have an impact on the final curve result, which is why the simulation solution swings so much. The

final test findings are consistent with the simulation results, demonstrating the accuracy of the model and the finite element simulation and providing assurance for further study.

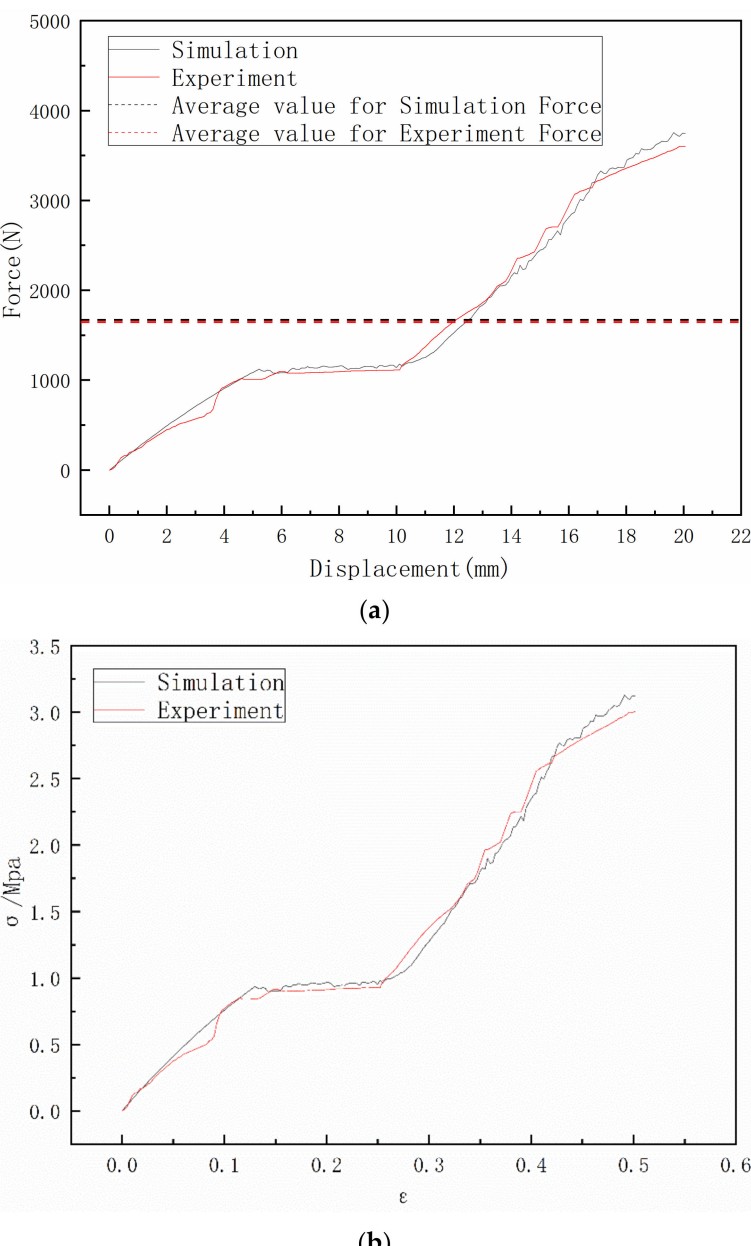

**Figure 11.** (**a**) Comparison of simulation results with experimental results. (**b**) Stress–strain curves of simulation results and experiment results.

*3.3. Results and Discussion*

3.3.1. Impact Resistance Indicators

Peak crushing force (PCF) and specific energy absorption (SEA) are chosen as crashworthiness indicators to measure the crashworthiness of cellular materials. PCF represents the maximum value along the impact process, which reflects the severity of the load on the cellular materials. SEA is often regarded as a common indicator for measuring the energy absorption capacity of cellular materials and the material utilization in energy absorption. The above crashworthiness indicators are defined as follows:

$$EA = \int_0^K F(\mathrm{x})dx, \tag{25}$$

$$PCF = \max[F(x)], \tag{26}$$

$$SEA = \frac{EA}{m}, \tag{27}$$

where $F(x)$ is the instantaneous crushing force, $K$ is the crushing distance when the material is compacted, $m$ is the total mass of cellular materials, and $EA$ denotes the energy absorbed via plastic deformation of cellular materials.

### 3.3.2. The Effect of Cellular Length $S$

To investigate the effect of the size of $S$ on the mechanical response of the concave I-shaped honeycomb sandwich structure under low velocity, five different sizes of $S$ were adopted in the numerical simulations, corresponding to group A in Table 1. The front view of single cellular structures of different sizes of $S$ is shown in Figure 12. The simulation results of the mass, SEA, and PCF in each case are listed in Table 2. In Figure 13a–c, the stress–strain, forces, energy, and specific energy absorption curves are compared for different sizes of $S$ in group A. Figure 13d shows the effect of the size of $S$ on specific energy absorption and peak crushing force. As shown in this figure, it can be concluded that, as the size of $S$ increases, both the specific energy absorption and peak crushing force curves decrease. Therefore, to some extent, increasing the size of $S$ can decrease the overall stiffness of the origami-inspired honeycomb sandwich plate.

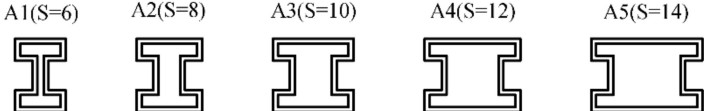

**Figure 12.** Cellular structures of different sizes of $S$.

**Table 2.** Numerical results of specimens with different sizes of $S$.

| Sample | Mass (g) | SEA (kJ/kg) | PCF (N) |
| --- | --- | --- | --- |
| A1 | 6.84 | 13.15 | 5921.96 |
| A2 | 7.55 | 8.79 | 4700.51 |
| A3 | 8.25 | 6.71 | 4051.05 |
| A4 | 8.95 | 5.14 | 3481.86 |
| A5 | 9.65 | 4.21 | 2766.76 |

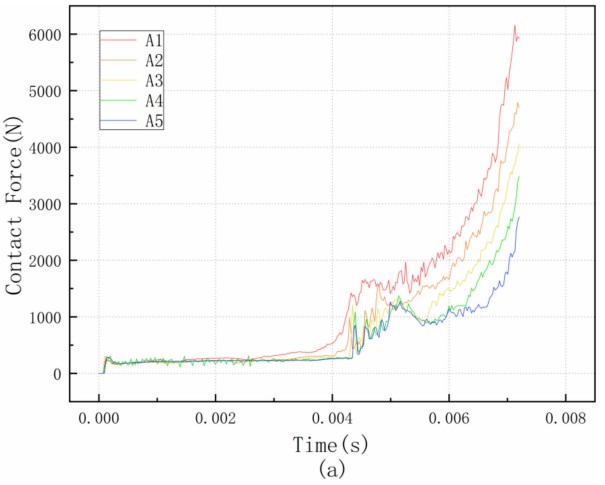

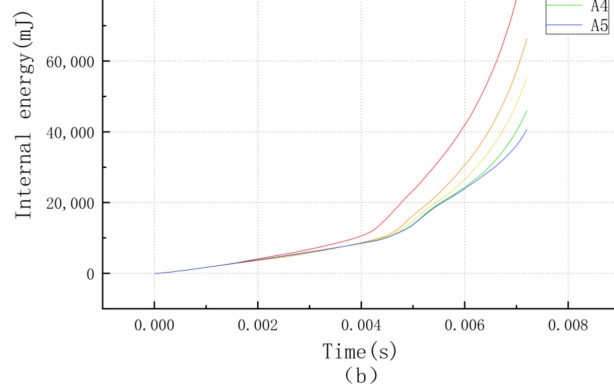

**Figure 13.** *Cont*.

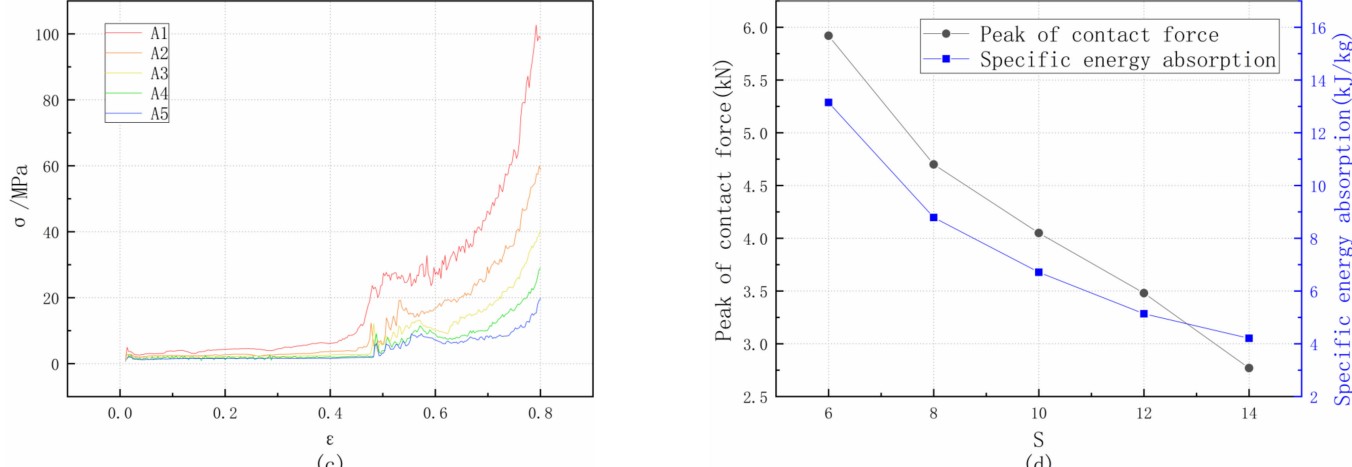

**Figure 13.** (**a**) Effect of different *S* on force; (**b**) effect of different *S* on energy; (**c**) effect of different *S* on stress–strain curve; (**d**) effect of different *S* on peak crushing force and specific absorption energy.

### 3.3.3. Effect of Vertical Short Cellular Height *h*

To investigate the effect of the size of *h* on the mechanical response of the concave I-shaped honeycomb sandwich structure under low velocity, five different sizes of *h* were adopted in the numerical simulations, corresponding to group B in Table 1. The front view of single cellular structures of different sizes of *h* is shown in Figure 14. The simulation results of the mass, SEA, and PCF in each case are listed in Table 3. In Figure 15a–c, the stress–strain, forces, energy, and specific energy absorption curves are compared for different sizes of *h* in group B. Figure 15d shows the effect of the size of *h* on specific energy absorption and peak crushing force. As shown in this figure, it can be concluded that as the size of *h* increases, both the specific energy absorption and peak crushing force curves decrease.

### 3.3.4. Effect of Horizontal Short Cellular Length *b*

To determine the effect of *b* on the response of the sandwich structure under the low-velocity impact, five different sizes of *b* (1.75 mm, 2 mm, 2.25 mm, 2.5 mm, and 2.75 mm) were used in simulations for the impact velocity of 10 m/s, with the parameters shown in Table 1, and these samples corresponded to group C. The single cellular structures of different sizes of *b* are shown in Figure 16. The curves of group C are shown in Figure 17: contact force (Figure 17a), internal energy (Figure 17b), stress–strain curves (Figure 17c), and the effect of *b* size on specific energy absorption and peak crushing force (Figure 17d). Additionally, detailed information on the mass, peak crushing force, and specific energy absorption of group C is listed in Table 4. It can be observed that the variation in the size of *b* has some effects on the mechanical property of the sandwich structure. The relationship between the size of *b* and the specific energy absorption and peak crushing force is presented in Figure 17d. As shown in this figure, it is confirmed that as the size of *b* increases, specific energy properties and peak crushing force increase.

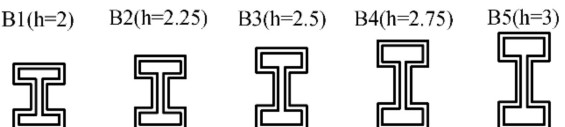

**Figure 14.** Cellular structures of different sizes of *h*.

**Table 3.** Numerical results of specimens with different sizes of *h*.

| Sample | Mass (g) | SEA (kJ/kg) | PCF (N) |
| --- | --- | --- | --- |
| B1 | 6.49 | 14.38 | 7923.09 |
| B2 | 6.84 | 13.15 | 5921.96 |
| B3 | 7.20 | 10.33 | 4285.29 |
| B4 | 7.55 | 7.68 | 2915.17 |
| B5 | 7.60 | 6.15 | 2722.18 |

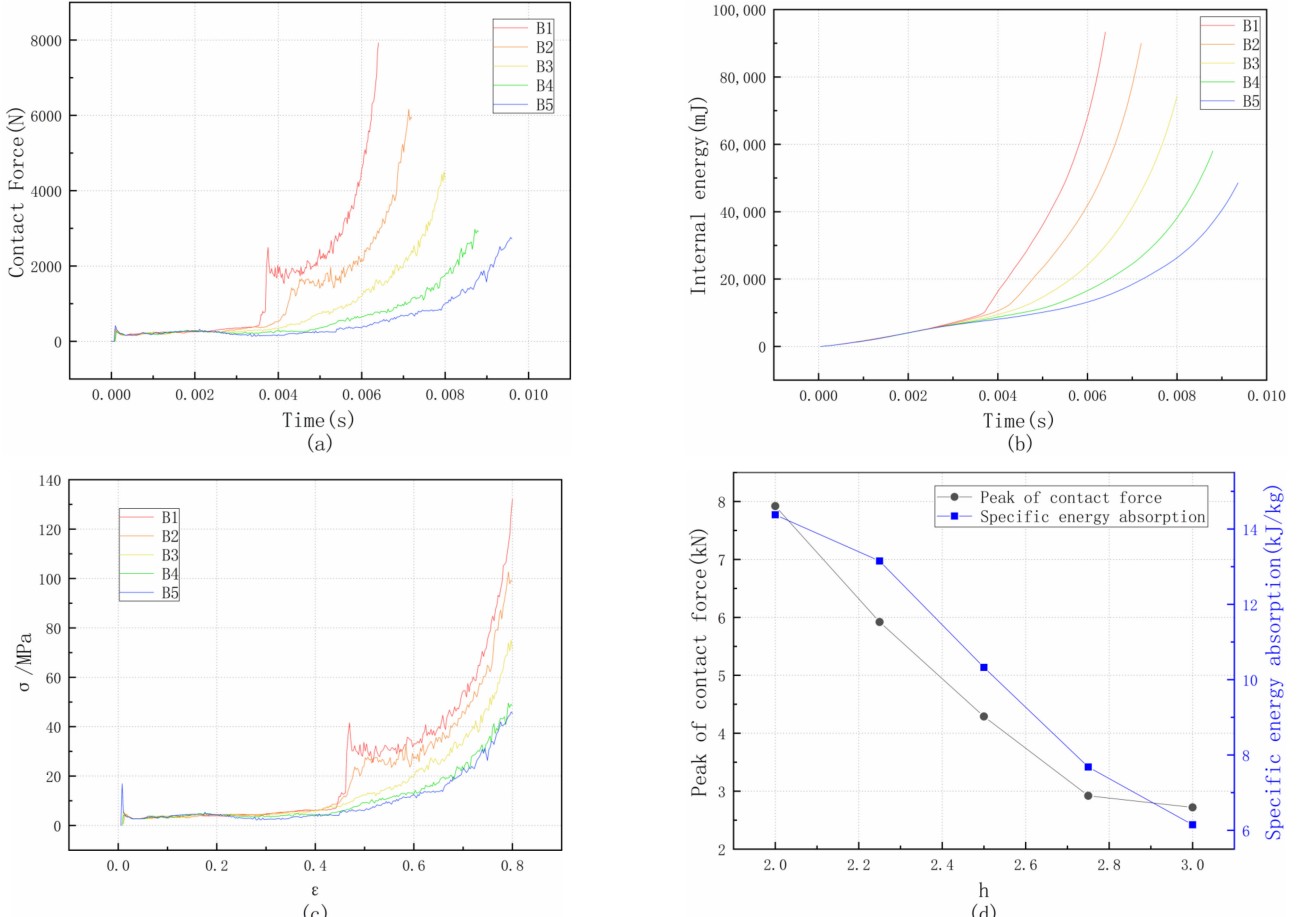

**Figure 15.** (**a**) Effect of different *h* on force; (**b**) effect of different *h* on energy; (**c**) effect of different *h* on stress–strain curve; (**d**) effect of different *h* on peak crushing force and specific absorption energy.

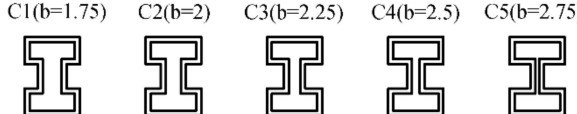

**Figure 16.** Cellular structures of different sizes *b*.

**Table 4.** Numerical results of specimens with different sizes of *b*.

| Sample | Mass (g) | SEA (kJ/kg) | PCF (N) |
| --- | --- | --- | --- |
| C1 | 6.49 | 10.26 | 4580.49 |
| C2 | 6.67 | 11.58 | 5539.27 |
| C3 | 6.84 | 13.15 | 5921.96 |
| C4 | 7.02 | 13.48 | 6290.71 |
| C5 | 7.20 | 13.69 | 6448.27 |

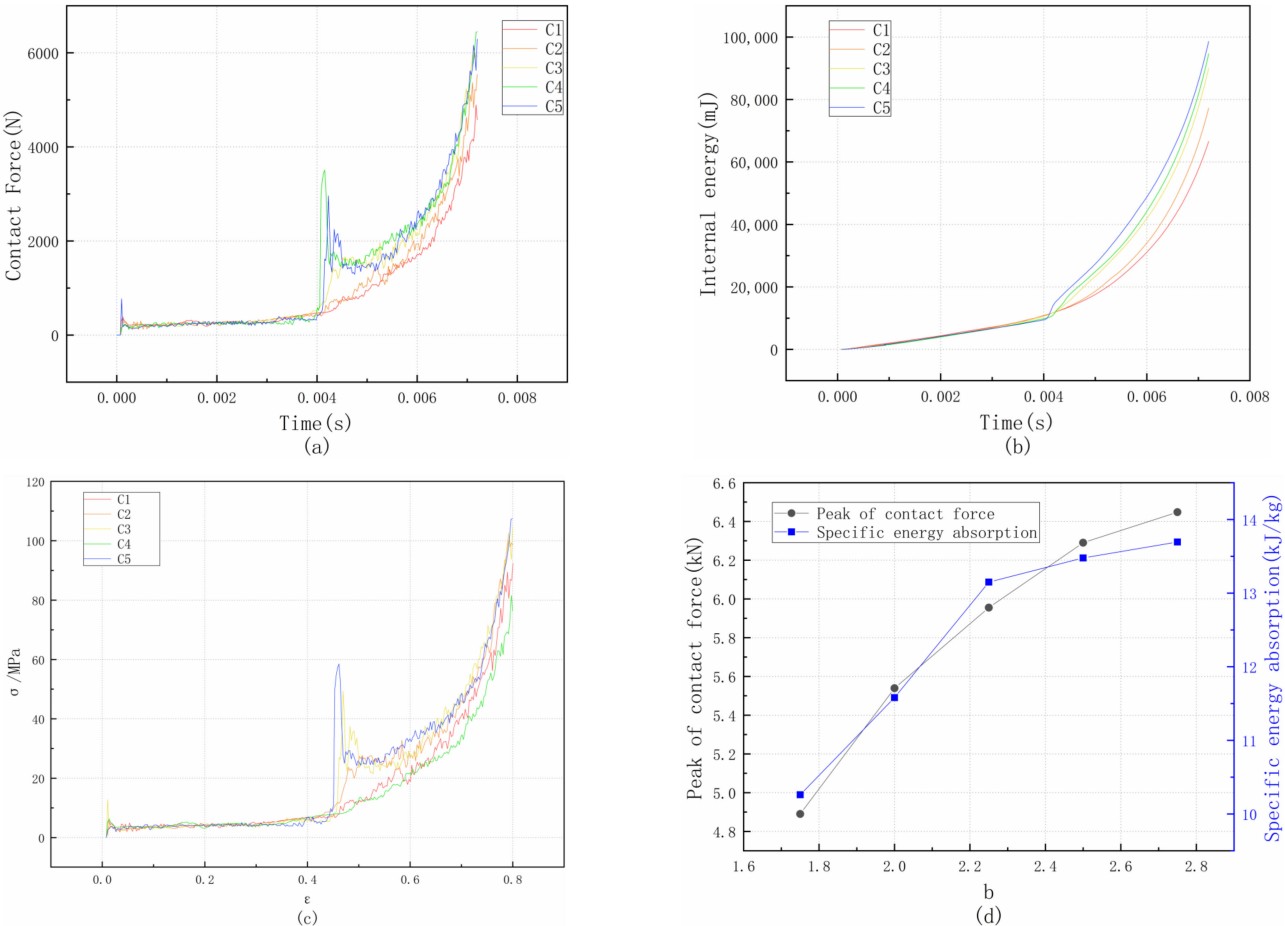

**Figure 17.** (**a**) Effect of different *b* on force; (**b**) effect of different *b* on energy; (**c**) effect of different *b* on stress–strain curve; (**d**) effect of different *b* on peak crushing force and specific absorption energy.

### 3.3.5. Effect of Cell Wall Thickness *t*

To determine the effect of *t* on the response of the sandwich structure under the low-velocity impact, five different sizes of *t* (0.5 mm, 0.65 mm, 0.8 mm, 0.95 mm, and 1.1 mm) were used in simulations for the impact velocity of 10 m/s, with the parameters shown in Table 1, and these samples corresponded to group D. The single cellular structures of different sizes of *t* are shown in Figure 18. The curves of group D are shown in Figure 19: contact force (Figure 19a), internal energy (Figure 19b), stress–strain curves (Figure 19c), and the effect of *t* size on specific energy absorption and peak crushing force (Figure 19d). Additionally, detailed information on the mass, peak crushing force, and specific energy absorption of group D is listed in Table 5. It can be observed that the variation in the size of *t* has some effects on the mechanical property of the sandwich structure. The relationship between the size of *t* and the specific energy absorption and peak crushing force is presented in Figure 19d. As shown in this figure, it is confirmed that as the size of *t* increases, specific energy properties and peak crushing force increase first and then start to drop.

D1(t=0.5)    D2(t=0.65)    D3(t=0.8)    D4(t=0.95)    D5(t=1.1)



**Figure 18.** Cellular structures of different sizes *t*.

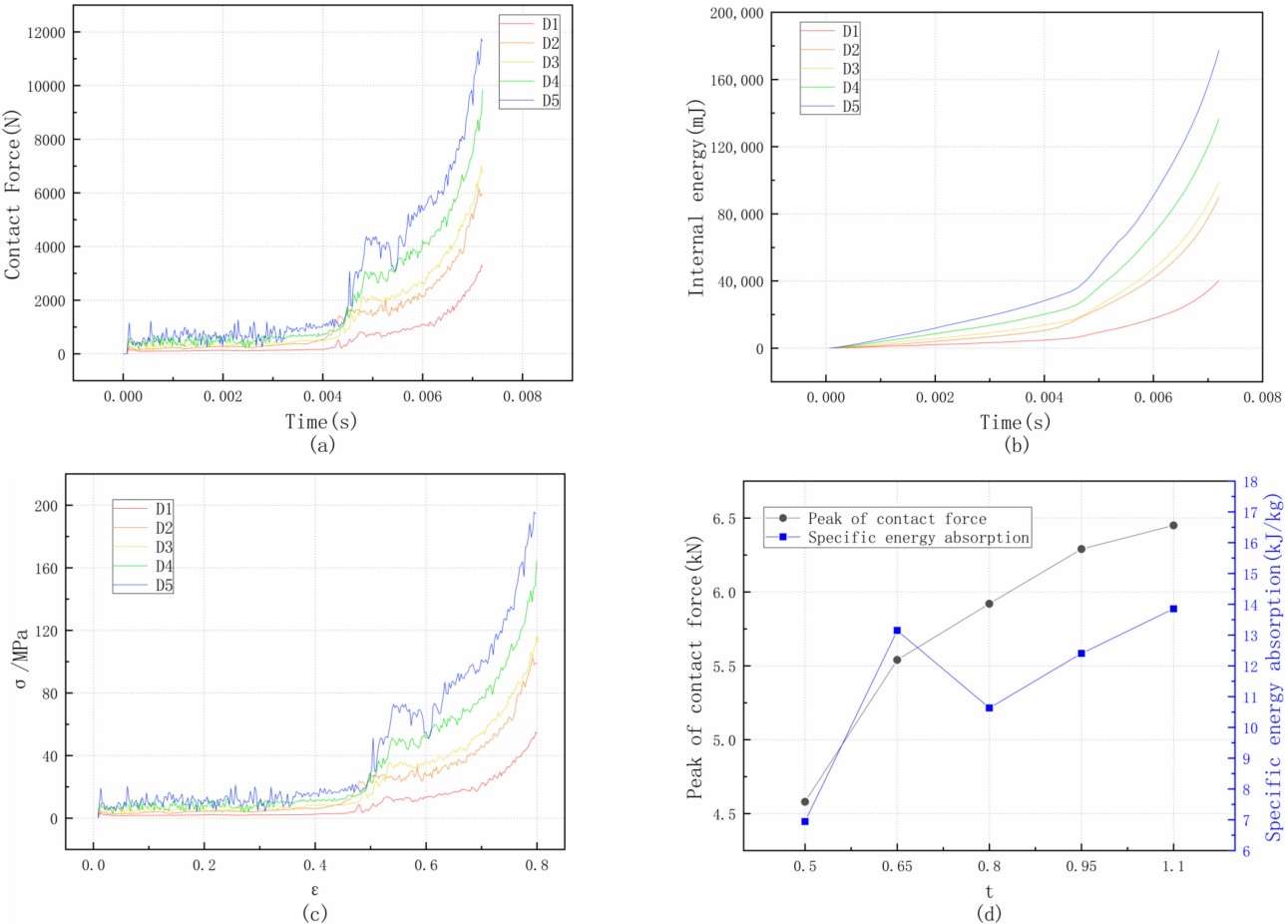

**Figure 19.** (**a**) Effect of different *t* on force; (**b**) effect of different *t* on energy; (**c**) effect of different *t* on stress–strain curve; (**d**) effect of different *t* on peak crushing force and specific absorption energy.

**Table 5.** Numerical results of specimens with different sizes of *t*.

| Sample | Mass (g) | SEA (kJ/kg) | PCF (N) |
|--------|----------|-------------|---------|
| D1 | 5.81 | 6.94 | 3256.83 |
| D2 | 6.84 | 13.15 | 5921.96 |
| D3 | 9.29 | 10.63 | 6744.81 |
| D4 | 11.00 | 12.40 | 9834.63 |
| D5 | 12.80 | 13.85 | 11,670.11 |

## 4. Multi-Objective Optimization of Concave I-Shaped Honeycomb Structure

The parameters of the concave I-shaped honeycomb structure influence the crashworthiness, which can be obtained from the analysis of the effect of the parameters of the cellular structure on crashworthiness in Section 3. Hence, the objectives, constraints, and design variables should be determined. Then, 30 groups of sample points are selected by Latin hypercube sampling, and the radial basis function model of different performance indexes is constructed according to the simulation results. Finally, the model of the optimization is built, and the optimum result of the concave I-shaped honeycomb structure is determined by the NSGA-II algorithm. The process of the multi-objective optimization method is depicted in Figure 20.

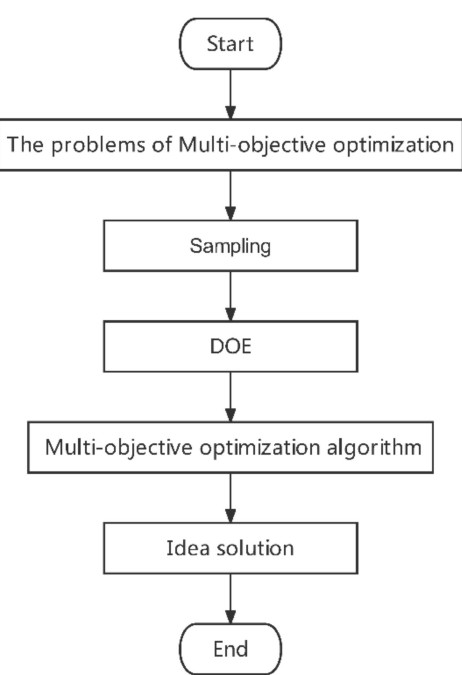

**Figure 20.** Multi-objective optimization workflow.

*4.1. Introduction to the Optimization Objectives*

The concave I-shaped honeycomb structure requires particular energy absorption and peak cellular force as the optimization targets to have greater crashworthiness. The idea of making lightweight materials is becoming more and more common nowadays. Mass is another objective of optimization, as people desire materials that are as light as feasible while still having a particular amount of impact resistance. Therefore, specific energy absorption, peak crushing force, and mass are the optimization objectives of this study. The cellular length $S$, the vertical short cellular height $h$, the horizontal short cellular length $b$, and the cellular thickness $t$ all influence the Poisson's ratio of the structure according to the theoretical calculation. Thus, $S, h, b$, and $t$ are selected as the variables. An appropriate range of parameters is chosen following Section 3 to guarantee the adaptability of the concave I-shaped honeycomb structure and the reasonableness of its macroscopic dimensions while taking into consideration the density and level of concavity of the honeycomb structure. The concave I-shaped honeycomb structure crashworthiness multi-objective problem is as follows:

$$
\begin{aligned}
&Min[-SEA, PCF, m]\\
&S_1 \leq S \leq S_2\\
&h_1 \leq h \leq h_2,\\
&b_1 \leq b \leq b_2\\
&t_1 \leq t \leq t_2
\end{aligned}
\tag{28}
$$

In Equation (28), $S_1$ and $S_2$ are the lower and upper limits of the cellular length and are taken as 6 mm and 14 mm, respectively; $h_1$ and $h_2$ are the lower and upper limits of the vertical short cellular height $h$ and are taken as 2 mm and 3 mm, respectively, $b_1$ and $b_2$ are the lower and upper limits of the horizontal short cellular length $b$ and are taken as 1.75 mm and 2.75 mm, respectively; and $t_1$ and $t_2$ are the lower and upper limits of the cellular thickness $t$ and taken as 0.5 mm and 1.1 mm, respectively.

*4.2. Selection of Sample Points*

The second stage of multi-objective optimization is the selection of sample points. The sampling method used should not only ensure that the data are accurate and representative but also ensure that the sample point coverage is consistent across the whole sample. Orthogonal test design [30], uniform design [31], and Latin square trials [32] are the

often used multi-objective optimization designs. The orthogonal test design has greater randomness but less uniformity; the homogeneous design has better uniformity but less randomness. As an extremely special multi-dimensional stratified sampling method, the Latin square test method is widely used in engineering, and it has both better uniformity and better randomness compared to the uniform and orthogonal test designs.

Latin hypercube sampling only uses one random sample point from each set, greatly preventing sample point collapse [33]. The basic idea is to randomly sample each variable by the interval after equally dividing each variable into M intervals per sample point in N dimensions. Additionally, it makes sure that the sampling space of the sample points is not duplicated. With fewer sample points, the technique enables a more precise description of the variable distribution. Thirty sets of sample points are chosen based on the range of optimization parameters, finite element simulations are run on them, and the results are integrated, as shown in Table 6.

**Table 6.** Thirty groups of sample points and response values.

| Sample | S (mm) | h (mm) | b (mm) | t (mm) | M (g) | SEA (kJ/kg) | PCF (kN) |
|--------|--------|--------|--------|--------|-------|-------------|----------|
| 1 | 8.665 | 2.950 | 2.279 | 0.878 | 11.90 | 14.71 | 9.91 |
| 2 | 10.008 | 2.112 | 2.240 | 1.086 | 13.50 | 23.31 | 16.73 |
| 3 | 11.606 | 2.277 | 1.954 | 0.598 | 7.95 | 10.41 | 5..61 |
| 4 | 7.508 | 2.708 | 2.539 | 0.971 | 12.30 | 21.86 | 11.55 |
| 5 | 10.484 | 2.355 | 2.705 | 0.990 | 13.50 | 19.26 | 10.45 |
| 6 | 11.751 | 2.492 | 2.625 | 0.931 | 13.61 | 15.21 | 9.51 |
| 7 | 8.885 | 2.877 | 2.370 | 0.525 | 7.13 | 11.52 | 5.00 |
| 8 | 9.438 | 2.401 | 1.919 | 0.736 | 9.09 | 15.13 | 8.99 |
| 9 | 11.258 | 2.990 | 1.818 | 0.678 | 9.83 | 9.09 | 5.63 |
| 10 | 6.659 | 2.332 | 1.789 | 0.685 | 7.24 | 18.57 | 7.69 |
| 11 | 9.762 | 2.016 | 2.31 | 1.013 | 12.30 | 25.14 | 19.33 |
| 12 | 9.175 | 2.513 | 2.451 | 0.551 | 7.18 | 12.14 | 4.59 |
| 13 | 10.967 | 2.827 | 2.116 | 0.644 | 9.22 | 10.61 | 6.43 |
| 14 | 6.121 | 2.568 | 2.088 | 0.806 | 8.95 | 19.13 | 8.98 |
| 15 | 8.251 | 2.616 | 2.010 | 0.574 | 7.05 | 14.04 | 5.66 |
| 16 | 7.012 | 2.064 | 2.415 | 0.748 | 8.12 | 25.21 | 14.22 |
| 17 | 10.647 | 2.772 | 2.591 | 1.055 | 15.30 | 15.60 | 9.88 |
| 18 | 8.369 | 2.266 | 2.566 | 0.604 | 7.36 | 16.14 | 8.08 |
| 19 | 10.396 | 2.126 | 2.354 | 1.042 | 13.30 | 22.08 | 17.14 |
| 20 | 6.635 | 2.876 | 1.859 | 0.966 | 11.40 | 17.19 | 9.52 |
| 21 | 9.739 | 2.431 | 1.931 | 0.735 | 9.26 | 14.33 | 9.27 |
| 22 | 9.955 | 2.387 | 2.141 | 1.093 | 14.00 | 20.34 | 14.90 |
| 23 | 10.888 | 2.538 | 2.537 | 0.880 | 12.40 | 13.18 | 8.13 |
| 24 | 11.416 | 2.919 | 2.488 | 1.009 | 15.30 | 14.20 | 10.47 |
| 25 | 11.987 | 2.153 | 2.086 | 0.909 | 12.20 | 15.54 | 12.99 |
| 26 | 7.818 | 2.499 | 2.698 | 1.011 | 12.70 | 24.97 | 12.92 |
| 27 | 10.559 | 2.558 | 2.201 | 0.682 | 9.28 | 10.93 | 5.81 |
| 28 | 11.311 | 2.060 | 2.607 | 0.760 | 10.20 | 16.78 | 12.64 |
| 29 | 6.331 | 2.210 | 1.794 | 0.536 | 5.43 | 16.14 | 5.24 |
| 30 | 9.218 | 2.622 | 2.340 | 0.587 | 7.73 | 12.17 | 4.70 |

### 4.3. Radial Basis Function Model (RBF) and Error Metrics

To streamline the multi-objective optimization process, it is required to determine how to fit an approximative model from the sampled points after the sampling issue has been resolved. The kriging model [34], the orthogonal polynomial model [35], the Gaussian radial basis function model [36], and the response surface model [37] are common fitting techniques used in data processing software. When using the kriging model, it is assumed that all data are unbiased and follow a normal distribution. Depending on the weight function, orthogonal polynomial models may be divided into Lejeune polynomials, Chebyshev polynomials, etc. A function may be fitted to the sum of radial basis functions using the Gaussian radial basis function, a real-valued function whose values depend

solely on the distance from the origin. This procedure may be compared to a basic neural network, which has the benefits of a simple structure, straightforward training, rapid learning convergence, the ability to approximate arbitrary nonlinear functions, and the ability to overcome local minimum values. Response surface models are fitted using polynomials, which are less efficient than neural network approaches in approximating extremely complicated functional connections but may be utilized to produce a more accurate model on a local scale with fewer sample points. The errors of the different approximation models are compared to choose an appropriate fitted model.

The R-squared ($R^2$), root mean squared error (RMSE), maximum absolute relative error (MARE), and average absolute relative error (AARE) are used to assess the approximation model's accuracy [38]. As demonstrated in Tables 7 and 8, the suitable proxy model approach is chosen based on the accuracy of the crashworthiness measures under each method.

$$R^2 = 1 - \frac{\sum\limits_{i=1}^{n} (y_i - \hat{y}_i)^2}{\sum\limits_{i=1}^{n} (y_i - \overline{y})^2}, \tag{29}$$

$$RMSE = \sqrt{\frac{\sum\limits_{i=1}^{n} (y_i - \hat{y}_i)^2}{n}}, \tag{30}$$

$$MARE = \max_{i=1,2,\dots,n} \left( \frac{|y_i - \hat{y}_i|}{|y_i|} \right), \tag{31}$$

$$AARE = \frac{1}{n} \sum\limits_{i=1}^{n} \frac{|y_i - \hat{y}_i|}{|y_i|}, \tag{32}$$

where $n$ is the number of test points, $y_i$ and $\hat{y}_i$ are the function value and the prediction value of the RBF model at the $i$th test point, and $\overline{y}$ represents the mean value of $y_i$.

**Table 7.** Accuracy assessment of the surrogate model.

| Response | Method | Accuracy Evaluation Coefficient | | | |
|----------|--------|--------|--------|--------|--------|
| | | $R^2$ | RMSE | MARE | AARE |
| M | Kriging | 0.93033 | 0.07312 | 0.31327 | 0.04141 |
| | OP | 0.14552 | 0.25607 | 1.11814 | 0.07808 |
| | RBF | 0.99878 | 0.00982 | 0.03629 | 0.00663 |
| SEA | Kriging | 0.86948 | 0.10178 | 0.33218 | 0.06905 |
| | OP | 0 | 5.88261 | 25.6863 | 1.79374 |
| | RBF | 0.94909 | 0.06341 | 0.13234 | 0.05146 |
| PCF | Kriging | 0.81532 | 0.11409 | 0.26187 | 0.09522 |
| | OP | 0 | 27.83448 | 121.5384 | 8.48736 |
| | RBF | 0.92577 | 0.07289 | 0.1382 | 0.06317 |

Generally, a larger value of $R^2$ means a more accurate metamodel. Smaller values of RMSE, MARE, and AARE correspond to a more accurate metamodel.

A parameter $T$ relating to the four error values is defined. When $T$ is smaller, it means that the error of the fitting method used is smaller.

$$T = (1 - R^2) + RMSE + MARE + RAAE, \tag{33}$$

$$
\begin{aligned}
T(Kriging) &= 1.78686 \\
T(OP) &= 195.5296 \\
T(RBF) &= 0.70057 \\
T(Order.1) &= 1.37095, \\
T(Order.2) &= 1.06566 \\
T(Order.3) &= 1.34652 \\
T(Order.4) &= 4.86487
\end{aligned}
\tag{34}
$$

Analysis of the comparative data shows that the RBF model has relatively small errors and is suitable for selection as an approximate model, so it is chosen as the surrogate model for subsequent calculations.

**Table 8.** Accuracy assessment of the surrogate model of response surface models 1–4.

| Response | Order | Accuracy Evaluation Coefficient | | | |
|---|---|---|---|---|---|
| | | $R^2$ | RMSE | MARE | AARE |
| M | 1 | 0.99127 | 0.02589 | 0.08527 | 0.0187 |
| | 2 | 0.99987 | 0.00318 | 0.00691 | 0.00243 |
| | 3 | 0.99975 | 0.00438 | 0.00979 | 0.00346 |
| | 4 | 0.99945 | 0.0065 | 0.01361 | 0.00522 |
| SEA | 1 | 0.87318 | 0.10032 | 0.20534 | 0.08042 |
| | 2 | 0.88697 | 0.09471 | 0.25124 | 0.07478 |
| | 3 | 0.9013 | 0.0885 | 0.21597 | 0.10662 |
| | 4 | 0.5665 | 0.18548 | 0.60132 | 0.1316 |
| PCF | 1 | 0.77801 | 0.12509 | 0.27177 | 0.10061 |
| | 2 | 0.87245 | 0.09482 | 0.21535 | 0.08153 |
| | 3 | 0.74157 | 0.13497 | 0.35768 | 0.06777 |
| | 4 | 0 | 0.40264 | 1.86361 | 0.22084 |

### 4.4. Presentation of NSGA-II

To solve the multi-objective optimization problem as formulated in Equation (28), the non-dominated sorting genetic algorithm II (NSGA-II) is used herein. It is an algorithm inspired by biological evolution and is a method for finding the optimal solution among all solutions. The non-dominated sorting genetic algorithm NSGA is an improvement of the genetic algorithm based on practical needs, and it has good results in solving multi-objective planning problems. The optimized second-generation non-dominated sorting genetic algorithm NSGA-II, compared with the previous generation algorithm, speeds up the computation in the optimal retention strategy and ensures population diversity in the genetic process [39]. The algorithm flow graph is shown in Figure 21. In this paper, the NSGA-II is used to solve the set of Pareto frontier solutions required by the multi-objective optimization design as a way to determine the optimal solution. The algorithm setup for NSGA-II is shown in Table 9.

**Table 9.** The general information on NSGA-II.

| Option | Value |
|---|---|
| Population Size | 20 |
| Number of Generations | 20 |
| Crossover Probability | 0.9 |
| Crossover Distribution Index | 10 |
| Mutation Distribution Index | 20 |

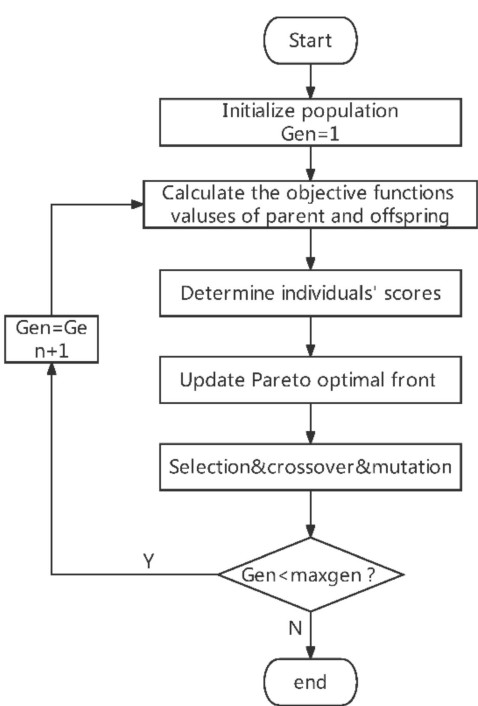

**Figure 21.** The optimization process of NSGA-II.

*4.5. Multi-Objective Optimization Results and Discussion*

The 401 sets of data are optimized, yielding 146 sets of workable optimum solutions. Figure 22 depicts the Pareto front distribution of SEA and PCF. SEA and PCF have a significant positive correlation, and as SEA steadily rises, so does PCF.

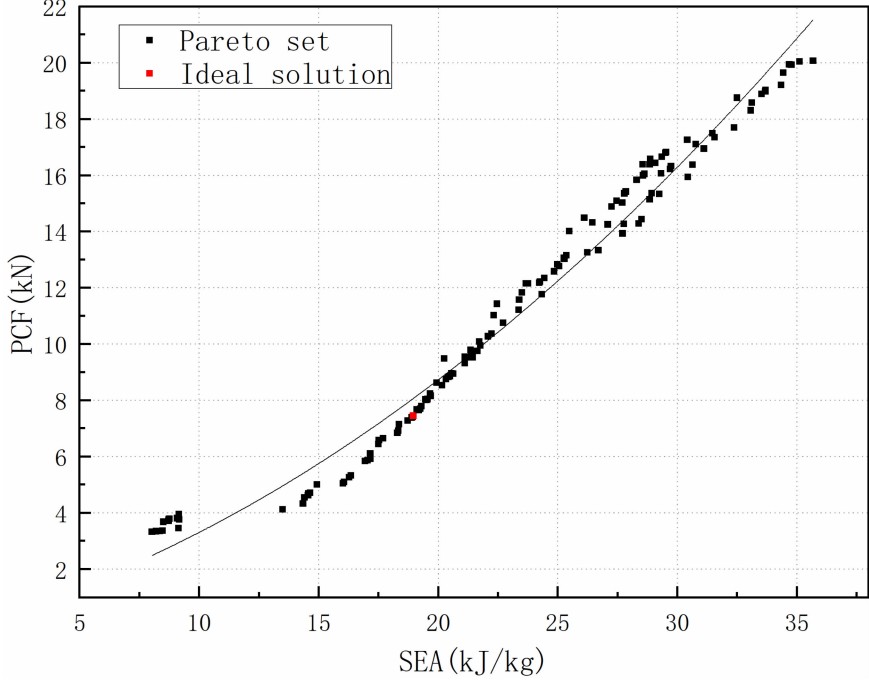

**Figure 22.** Pareto fronts from the surrogate model.

The design objectives SEA, PCF, and mass are 22.196 kJ/kg, 7.433 kN, and 7.777 g, respectively. The values are $S$ = 6.026 mm, $h$ = 2.0 mm, $b$ = 2.039 mm, and $t$ = 0.501 mm for each optimization parameter.

According to the values of the optimized parameters, the corresponding finite element simulation model is established, finite element simulation is carried out, and the results are compared with the optimized results, as shown in Table 10. The inaccuracy is within a manageable range, and it can be confirmed that the concave I-shaped honeycomb structure approximation optimization model's optimization approach is correct.

**Table 10.** Comparison between simulation and KBF model of optimum point.

|  | Mass (g) | SEA (kJ/kg) | PCF (kN) |
|---|---|---|---|
| KBF model | 4.710 | 18.959 | 7.452 |
| Simulation | 4.890 | 18.615 | 7.382 |
| Error | 3.822% | 1.814% | 0.939% |

Figure 23 and Table 11 provide a comparison of the original solutions with the optimized results. It can be seen that although the mass has been significantly lowered and the peak crushing force has risen within a reasonable range, the specific energy absorption has been much enhanced, demonstrating the effectiveness of optimization.

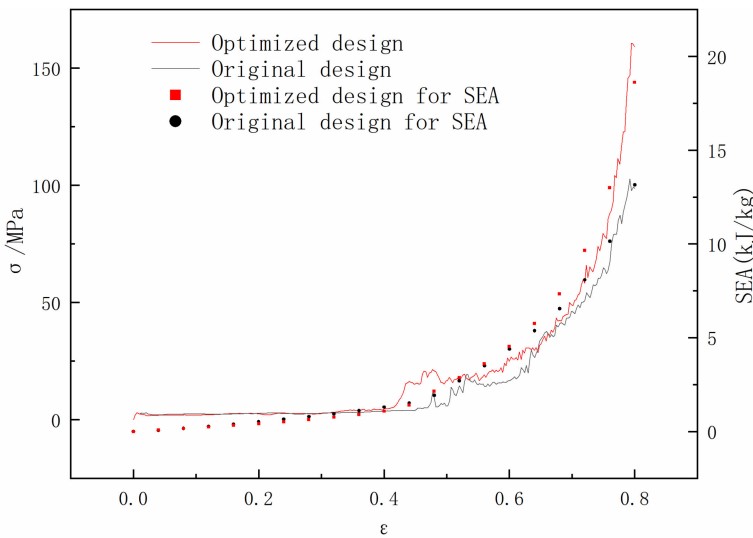

**Figure 23.** Comparison of stress–strain between optimized and original results.

**Table 11.** Comparison of optimized and original results.

|  | Mass (g) | SEA (kJ/kg) | PCF (kN) |
|---|---|---|---|
| Initial result | 6.840 | 13.150 | 5.921 |
| Optimized result | 4.710 | 18.959 | 7.452 |
| Change | −31.140% | 44.175% | 25.857% |

## 5. Conclusions

In this paper, the study of the concave I-shaped honeycomb structure under axial loading was performed with experiment and finite element analysis methods. Based on the validated finite element methods, the four parameters were proposed and analyzed to improve crashworthiness. To obtain the optimum designs of the structure, NSGA-II was used with the variables $S$, $h$, $b$, and t as well as objective functions of SEA, PCF, and mass. The following are the summarized conclusions resulting from this study:

- The combination of the traditional re-entrant hexagon structure and the I-beam structure produces the concave I-shaped honeycomb structure. The effect of each parameter on negative Poisson's ratio is investigated using the energy method. The Poisson's ratio of the structure is increased with an increase cellular length $S$, and it is decreased

         with decreases in the vertical short cell wall height $h$ and the horizontal short cell wall $b$. The structure has negative Poisson's ratio characteristics at certain dimensions.

- The cellular length $S$, the vertical short cellular length $h$, the horizontal short cellular length $b$, and the cellular thickness $t$ are defined in the cellular structure, and the effects of these four parameters on the specific absorption energy and the peak crushing force of the impact resistance index of the cellular structure are investigated. The results show that the values of specific absorption energy and peak crushing force decrease with increases in the variables $S$ and $h$, and they increase with increases in the variables $b$ and $t$, which are all factors affecting the impact resistance of the cellular structure.

- The specific absorption energy and peak crushing force are found to be positively correlated by optimizing the design. The geometric parameters of the optimal structure are $S = 6.026$ mm, $h = 2.0$ mm, $b = 2.039$ mm, and $t = 0.501$ mm. The values of specific absorbed energy, peak force, and mass are 18.959 kJ/kg, 7.452 kN, and 4.710 g. Compared with the original design, the SEA increased by 44.175%, and the PCF increased by 25.857%. Meanwhile, the mass decreased by 31.140%. Hence, the optimal structure has better crashworthiness.

**Author Contributions:** Methodology, M.L.; software, M.L.; validation, T.W. and M.L.; formal analysis, D.Q.; writing—original draft preparation, M.L.; writing—review and editing, T.W.; visualization, T.W.; supervision, D.Q.; project administration, J.C. and H.W.; funding acquisition, T.W. All authors have read and agreed to the published version of the manuscript.

**Funding:** This work was supported by the Nature Science Foundation of China, funding number 51705468 and the Key Technologies Research and Development Program, funding number 2018YFB0106204.

**Institutional Review Board Statement:** Not applicable.

**Informed Consent Statement:** Not applicable.

**Data Availability Statement:** Not applicable.

**Conflicts of Interest:** The authors declare no conflict of interest. The funders had no role in the design of the study; in the collection, analyses, or interpretation of data; in the writing of the manuscript; or in the decision to publish the results.

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
