# Peer review of "Crashworthiness Analysis and Multi-Objective Optimization for Concave I-Shaped Honeycomb Structure"

_applsci, doi:10.3390/app122010420_

Round 1

Reviewer 2 Report

In this manuscript, Wang et al. described crashworthiness analysis and multi-objective optimization for 2 concave I-shaped honeycomb structure. The author's stated that materials with negative Poisson's ratio are of increasing interest to research scholars, especially in fuel-efficient vehicles, due to their superior structural and mechanical properties. Thus, in the study they perfomed the concave I-shaped honeycomb structure under axial loading with experiment and finite element analysis methods. Based on the validated finite element methods, they proposed the four parameters and analyzed to improve the crashworthiness. Next, to obtain the optimum designs of the structure, they used the variables S, h, b, and t as well as objective functions of SEA, PCF, and mass with NSGA-II.

From my point of view, the manuscript is interesting and will attract the interest of many scientists working with such systems. However, the manuscript requires careful editing as there are some minor typos and other flaws. Moreover, extensive editing of English language and style is required.

Reviewer 3 Report

·       Is important to add numerical data and results to the abstract. You need to highlight your best results and your contribution.

·       The authors propose a very interesting analysis with for industrially importance in the metamaterials research. Although, the honeycomb structure is highly studied it has a high scientific interest.

·       In the introduction the authors explain very well the state of the art of the cellular structures but not about the manufacturing process.

·       The simulation and mathematical model is very well explained, but the experimental validation is missing very critical information.

·       I highly recommend explaining with more detailed the experimental validation. Add the model and technique of the 3d printed, the material used for the samples, printing parameters, etc.

·       Several spelling and grammar errors in all the manuscript.

Reviewer 4 Report

The work was well explained and easy to follow. I would suggest a few minor things.

1. It would be better if the authors checked the fonts and the clarity of the figures and figure captions.

2. It would be better if the font size in the flow diagram was increased a bit.

3. Why was the conclusion written as bullet points?

4. It would be better if the section titles were revisited and rephrased.

5. What is the novelty of this work?

Round 2

Reviewer 3 Report

I want to thank the authors for the time and dedication to make the changes that were requested.